# Mechano-regulated metal–organic framework nanofilm for ultrasensitive and anti-jamming strain sensing

Liang Pan[1,2,3], Gang Liu[1,3], Wenxiong Shi[2], Jie Shang[1,3], Wan Ru Leow[2], Yaqing Liu[2],
Ying Jiang[2], Shuzhou Li[2], Xiaodong Chen[2] & Run-Wei Li[1,3]

The development of ultrasensitive, anti-jamming, and durable sensors that can precisely distinguish different human body motions are of great importance for smart health monitoring and diagnosis. Physical implementation of such flexible sensors is still a challenge at the moment. Combining the designs of advanced material showing excellent electrochemical properties with the facilitative structure engineering, high-performance flexible sensors that satisfy both signal detecting and recognition requirements may be made possible. Here we report the first metal–organic framework-based strain sensor with accurate signal detection and noise-screening properties. Upon doping the tricarboxytriphenyl amine-based metal–organic framework nanofilm with iodine, the two-terminal device exhibits ultrahigh sensitivity with a gauge factor exceeding 10,000 in the 2.5% to 3.3% deformation range for over 5000 dynamic operating cycles and out-of-scale noise-screening capability. The high-performance strain sensor can easily differentiate the moderate muscle hyperspasmia from subtle swaying and vigorous sporting activities.

---

[1] CAS Key Laboratory of Magnetic Materials and Devices, Ningbo Institute of Materials Technology and Engineering, Chinese Academy of Sciences, Ningbo, Zhejiang 315201, China. [2] School of Materials Science and Engineering, Nanyang Technological University, 50 Nanyang Avenue, 639798 Singapore, Singapore. [3] Zhejiang Province Key Laboratory of Magnetic Materials and Application Technology, Ningbo Institute of Materials Technology and Engineering, Chinese Academy of Sciences, Ningbo 315201, China. Correspondence and requests for materials should be addressed to G.L. (email: liug@nimte.ac.cn) or to X.C. (email: chenxd@ntu.edu.sg) or to R.-W.L. (email: runweili@nimte.ac.cn)

Keeping pace with the recently fast-developed wearable personalized health-monitoring equipments, flexible strain sensors that can translate the gentle human body motion into electric capacitance or resistance changes in an ultrasensitive and anti-jamming manner are of great medication importance for elder and patient communities[1–5]. For instance, accurate discrimination of muscle hyperspasmia from anoetic swaying and vigorous sporting activities makes real-time tracking of epilepsy possible via remote online diagnosis with flexible strain sensor, allowing the delivery of timely healthcare service to patients at all time and places[6]. This raises two basic sensing requirements: signal detecting and recognition[7–9]. With the direct disruption and recovery of conducting pathways by tuning the width of nanogap[10–13], crack-type sensors guarantee the highest sensitivity so far for signal detecting among the existing geometrical and piezoresistive strain-responding mechanisms[14–16]. Nevertheless, the indiscriminate gauging of subtle vibration at its low detecting limit ($\varepsilon < 1\%$) makes the noise-screening criteria self-contradictory[17], whereas the uncontrolled evolution of nano-cracks over the entire sensing film inevitably leads to severe deterioration in the device reliability[18]. Physical implementation of the ultrasensitive, anti-jamming, and durable flexible strain sensor is still a challenge at the moment.

Marrying the designs of advanced sensing materials carrying superior intrinsic electromechanical feature with facilitative structure engineering may offer promising opportunities of simultaneously satisfying the signal detecting and recognition requirements[19]. Metal–organic frameworks (MOFs) are crystalline nanoporous materials with supermolecular structures consisting of metal cations (or clusters) and soft organic bridging ligands[20,21]. The mature and rich coordination chemistry enables a versatile synthetic platform for the bottom-up design of MOF materials with anticipated geometries and properties, and directly determines their industrial applications through specific building units and host–guest interactions[22–28]. It is noteworthy that both the metallic nodes and organic ligands may serve as carrier transport sites via through-bond or through-space conduction in the crystal lattices[29–33], frameworks with large bridging linkers, and moderate bond strength, and thus signature deformable nanoporosity would allow effective modulation of the inter-atom spacing between the charge-transporting spots under mechanical stress and make flexible MOFs reliable molecular analog to the crack-based strain sensors[34–38]. Signals stemming through local deformation can be further amplified by network cascading along the three-dimensional (3D) MOF skeleton with short- and long-range ordering, allowing ultrasensitive detecting of tiny strain with these hybrid materials[39].

Herein, we report the first flexible $I_2$@CuTCA ($I_2$@$Cu_3(C_{21}H_{12}NO_6)_2$) MOF nanofilm-based strain sensor that fulfills the ultrasensitive and anti-jamming criteria for accurate human body motion recognition. $I_2$@CuTCA nanofilm was synthesized by chelating redox active tricarboxytriphenyl amine ($H_3$TCA) ligand with divalent $Cu^{2+}$ cations to create the soft skeleton and doping with iodine to form semiconductive charge-transfer (CT) complex. Upon effective modulation of the charge-carrier hopping between adjacent $I_2$-TCA complexes by the strain-induced shrinkage of the framework in the out-of-plane direction, superior gauge factor of as high as 11,120 is observed at the strain level of 3.3%, together with the durability of exceeding 5000 dynamic operating cycles achieved in the strain range between 2.5% and 3.3%. Interestingly, integrating $I_2$@CuTCA into a vertical Au/MOF/Au sandwich structure can release the out-of-scale strain through grain boundaries of the particulate nanofilm readily, giving rise to non-responsiveness at strain levels lower than 2.5% or higher than 3.3%, and, more importantly, the middle-range responsive performance for the

fine recognition of human body motions. As such, a smart kneecap that can accurately distinguish the leg movements of subtle swaying, walking, and vigorous bicycling is demonstrated.

## Results

**MOF design and synthesis.** MOFs distinguish themselves with ordered structures defined by the coordination geometry of the metal nodes and topology of the organic bridging linkers[28]. On one hand, the 3D long-range ordering of the repetitive building blocks may provide potential charge transport pathways with either through-bond or through-space conduction model; on the other hand, the mechanical flexibility of the organic–inorganic hybrids puts strain detection into a possibility by the deformation-induced electronic structure and conductivity change of the material. However, the low atomic density and strong localization of the electron wave function of MOFs lead to scarcity in the itinerant electrons and electronic communication between the metal cations and organic ligands, resulting in insulating behavior that can hardly be used for electronic devices[39–41]. Increasing MOFs' conductivity to semiconductive level thus appears a direct necessity for device applications. With this concern, we use the redox active $H_3$TCA ($H_3$TCA = $C_{21}H_{15}NO_6$) entities as the bridging ligands to chelate divalent $Cu^{2+}$ cations (Fig. 1a), not only aiming to improve porosity and flexibility of the CuTCA ($Cu_3(C_{21}H_{12}NO_6)_2$) framework but also generating CT complex with the guest iodine molecules to enhance the charge transport efficiency[42–44]. Herein, the nitrogen atom with lone-pair electrons in the center of the propeller-shaped $H_3$TCA ligand can pin the $I_2$ molecules tightly to the framework and increase the carrier concentration through oxidative doping.

Pristine CuTCA nanofilms were first prepared on gold/titanium/polyethylene terephthalate (PET) flexible substrates through a modified liquid-phase epitaxy (LPE) approach developed by our group[45–47]. The as-fabricated CuTCA film has a particulate nature, showing a typical thickness of ~100 nm and promising morphological uniformity with the root mean square roughness of ~3 nm and maximum peak-to-peak height of around 10 nm (Fig. 1b). The lateral dimension of the CuTCA nanofilm is 1 cm × 1 cm as determined by that of the Au/PET substrate, and can be cut into arbitrary shapes and sizes at will. X-ray diffraction (XRD) pattern suggests that the CuTCA film is (111) direction-oriented (Fig. 1c), which is similar to the well-known fcc-structured HKUST-1 and in good agreement with that reported in the literatures[21,24,48]. The CuTCA unit cell is 23.211 Å in all the a, b, and c directions, wherein the use of relative large TCA linker allows sufficient free space for guest molecule infiltration. Current–voltage (I–V) characteristics of the CuTCA film was monitored in an Au/MOF/Au sandwich structure, which shows low conductance of ~$10^{-9}$ S and the as-expected insulating nature (Fig. 1e). The MOF devices are 380 nm in thickness (the top and bottom Au electrodes are 80 nm and 200 nm in thickness, respectively) and 100 μm in diameter. The CuTCA film was also infiltrated with iodine, by first heating the sample under reduced pressure at 100 °C for 30 min, to remove the residue solvent molecules, then immediately immersing in a 2 mL $I_2$/ethanol solution with the iodine concentration of 100 μM for 48 h. Afterwards, the sample was heated at 80 °C for 1~2 h, to remove the residue solvent and free iodine molecules that do not form stable interaction with the $H_3$TCA ligand. Doping CuTCA with excess iodine can guarantee an at least 1:1 ratio of $I_2$ molecule/N atom inside the framework, thus producing sufficient amount of charge carriers and sample conductivity for signal detecting. The color of the MOF film changes from dark green of pure CuTCA to purple of $I_2$@CuTCA, which is in close similarity to that of iodine. The surface morphology of the MOF film is well

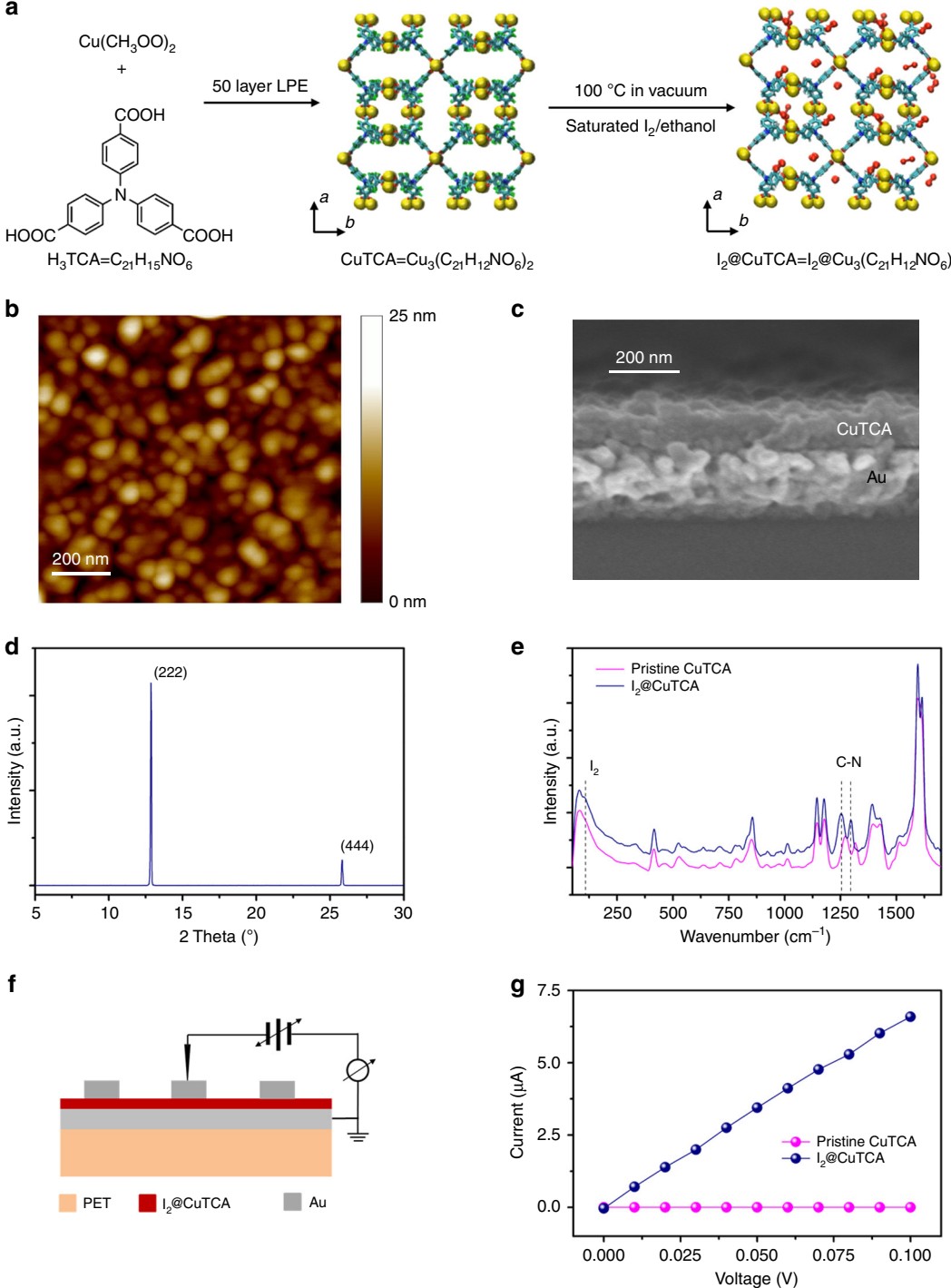

**Fig. 1** Liquid-phase epitaxial synthesis (LPE) of CuTCA and $I_2$@CuTCA nanofilms. **a** Synthetic route of the metal–organic framework (MOF) nanofilms. The chemical formula of $H_3$TCA, CuTCA, and $I_2$@CuTCA are $C_{21}H_{15}NO_6$, $Cu_3(C_{21}H_{12}NO_6)_2$, and $I_2$@ $Cu_3(C_{21}H_{12}NO_6)_2$, respectively. The cyan, white, blue, and red twigs represent carbon, hydrogen, nitrogen, and oxygen atoms, respectively. The yellow and red spheres stand for copper and iodine atoms, respectively. **b** Atomic force microscopic top-view image and **c** scanning electronic microscopic cross-section view image of the CuTCA nanofilm deposited on Au/PET (polyethylene terephthalate) substrate. **d** X-ray diffractive pattern of the CuTCA nanofilm. **e** Raman spectra of the pristine CuTCA and iodine-doped $I_2$@CuTCA nanofilms. **f** Schematic illustration of the Au/$I_2$@CuTCA/Au structure on PET substrate for electrical measurements. **g** Current–voltage characteristics of the pristine CuTCA and iodine-doped $I_2$@CuTCA nanofilms

maintained upon doping (Supplementary Figure 1). Raman spectra of the iodine-doped MOF shows that the vibration stretching peaks of the C–N bonds shift from the wavenumbers of 1273 cm$^{-1}$ and 1317 cm$^{-1}$ to 1251 cm$^{-1}$ and 1295 cm$^{-1}$, respectively, whereas a new peak associated with the adsorbed $I_2$ molecules appears at 106 cm$^{-1}$ (Fig. 1d and Supplementary Note 1). Core-level N 1 s X-ray photoelectron spectrum (XPS) of the $I_2$-doped CuTCA nanofilm also reveals noticeable appearance

of new nitrogen species with the binding energies (BEs) of ~399.2 eV and 400.2 eV, which can be ascribed to the presence of neutral N atoms at the higher oxidative state (e.g., imine) and the positively charged N atoms, respectively (Supplementary Figure 2a, b and Supplementary Note 2). The XPS spectral area ratio of the iodine and nitrogen species in the $I_2$@CuTCA nanofilm suggests that an as-expected 1.07:1 ratio of the $I_2$ molecule/N atom is obtained (Supplementary Figure 2c, d). Therefore, the observed changes in the lineshape of the Raman and XPS spectra upon doping CuTCA with $I_2$ strongly illustrate that CT interaction occurs between iodine and the framework,[30,49–51] resulting in a $10^5$ fold of increase in the conductance of the Au/MOF/Au device (Fig. 1e). The conductivity of the iodine-doped CuTCA is ~$1.3 \times 10^{-3}$ S/m, which is similar to that of the high-mobility organic semiconductors[52].

**Molecular dynamics simulation.** In order to assess the strain detecting possibility of using MOF materials as the sensing media, molecular dynamics (MD) simulation was performed to visualize the lattice geometry evolution and corresponding electronic structure change of the $I_2$@ CuTCA crystallite under different strains (Supplementary Note 3). A $2 \times 2 \times 2$ super cell of CuTCA containing 64 infiltrated $I_2$ molecules was employed for theoretical calculation, wherein a 1:1 ratio of the $I_2$ molecule/N atom was maintained in accordance with the sample composition. In general, ordinary materials usually demonstrate positive Poisson's ratios; thus, the introduction of tensile strain in one direction will result in shrinkage of the materials on their cross sectional areas. As the thickness of the PET substrate (~175 μm) is much thicker than that of the Au/$I_2$@CuTCA/Au device (~380 nm), the neutral plane of the entire sample is located in the PET substrate.

Bending such samples upwards will result in in-plane tensile (stretching) strain and thus shrinkage of the MOF layer in the out-of-plane direction, which in turn translate the mechanical deformation into change in vertical electrical signals. Accordingly, we evaluate the cross-section geometry and electronic structure variation of the $I_2$@CuTCA framework during gradual shrinkage processes as shown in Fig. 2a.

With the limited electronic coupling between the copper nodes and TCA linkers, charge carriers are locally trapped at the lattice sites and the through-bond conduction is greatly restrained in the pristine CuTCA framework. The relatively large distance between the adjacent TCA linkers (with the shortest inter-N atom distance of 11.60 Å) also depresses the possibility of through-space conduction via carrier delocalization[36]. As such, the as-prepared CuTCA film is insulating with ignorable current leakage at the grain boundaries (Supplementary Figure 3). Upon infiltration, the guest iodine molecules only exhibit minor vibration around the nitrogen atoms of the TCA ligands, instead of traveling erratically inside the free MOF pore-space in a Brownian mode (Supplementary Movie 1). The distance between iodine and TCA linkers (1.59 Å) is smaller than the van der Waals distance (Fig. 2b), which allows CT interaction to occur between them. The formation of the $I_2$-TCA CT complex can move the Fermi level toward the edge of the valance band, leading to oxidative doping of the framework with increased hole concentration in the bandgap and greatly enhanced conductivity, as well as potential responsiveness to mechanical stimuli[23,42,43]. Owing to the large size of the TCA linker and small size of the $I_2$ molecule, continuous through-bond conduction pathway may not be established by CT interaction, as in the case of TCNQ@HKUST-1[30]. The unaffected XRD pattern of $I_2$@CuTCA without any newly appearing Bragg peaks also suggests the lack of

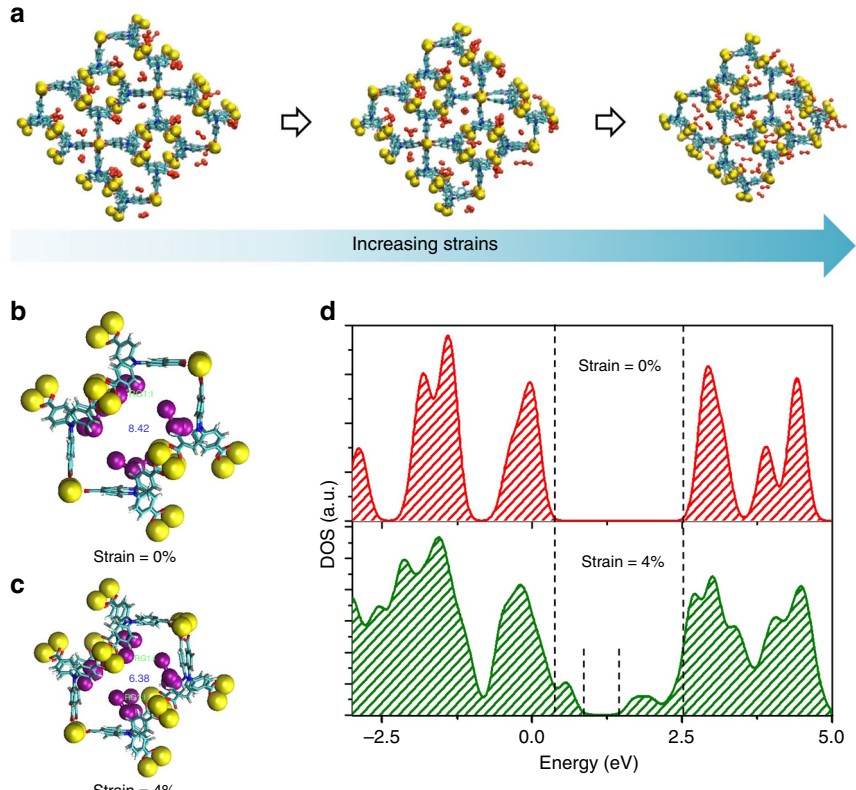

**Fig. 2** Evolution of the crystalline and electronic structures of $I_2$@CuTCA. **a** Shrinkage of the $I_2$@CuTCA framework under increasing strains. **b, c** Inter-atom distances between the neighboring $I_2$ hopping sites in $I_2$@CuTCA under the bending strains of 0 and 4%, respectively. In **b**, the iodine atoms are highlighted in purple spheres for a better illustration. **d** Local density of states (DOS) of $I_2$@CuTCA under the bending strain of 0% and 4%, respectively

long-range ordered conduction pathway in the iodine-doped MOF framework (Supplementary Figure 4a). On the other hand, the distance between the adjacent $I_2$-TCA complexes (8.42 Å) is still larger than the van der Waals distance, indicating that through-space band conduction is unlikely to occur[53,54]. Nevertheless, the temperature-dependent $I$–$V$ measurement of the $I_2$@CuTCA device shows thermally activated conductivity (Supplementary Figure 4b), suggesting that hopping mechanism may account for its semiconductive nature. When the MOF framework shrinks, the $I_2$-TCA to $I_2$-TCA distance decreases gradually and reaches 6.38 Å at the strain level of 4% (Fig. 2c). The shortened distance between the hopping sites can thus increase the hopping probability and consequently the charge-carrier mobility. An accompanying lowering of the energy bandgap of 1.49 eV upon deformation is also demonstrated, which can be ascribed to the enhanced electronic communication between the adjacent $I_2$-TCA CT complexes (Fig. 2d)[53]. Consequently, the local deformation of MOF lattice in response to external strains would enhance the charge transport across the nanofilm and engender strain-responsive conductivity of the Au/ $I_2$@CuTCA/Au device[39,55].

**Robust strain detection and noise screening.** Due to the mismatch between the lattice constants of the gold-bottom electrode and the MOF materials, as well as the fast film growth rate with the modified LPE approach, the as-fabricated CuTCA nanofilm has a preferentially oriented crystalline particulate rather than single crystal nature. When integrated in an Au/$I_2$@CuTCA/Au structure with device current flowing vertically between the top and bottom electrodes, the grain boundaries of the MOF film may release the horizontal strains generated upon being bent to either a very small or a very large extent, thus leading to non-responsive conductance of the device in these scenarios. This is different from the traditional planar structure strain sensors that

monitors the lateral device currents, and helps to screen the environmental noises and uninterested violent deformations. The origin of the out-of-scale non-responsiveness of the device will be discussed later. The electrical behavior of the vertically stacked Au/$I_2$@CuTCA/Au sensor device fabricated on flexible PET substrate was evaluated with a home-made apparatus as shown in Fig. 3, Supplementary Figures 5, 6, which can bend the sample to different levels quantitatively (Supplementary Note 4). The virgin devices are always in low conductance states of ~$1.67 \times 10^{-5}$ S (Fig. 3a). In good agreement with our proposal, the $I$–$V$ curves of the devices almost overlap with each other when the applied bending strain increases from 0% to 2.5%, suggesting that the conductivity of the MOF layer remains unchanged at low deformation levels. The device currents increase rapidly as the strain ramps from 2.5% to 3.3%, which may be attributed to the molecular-scale sensing via enhanced carrier hopping between the neighboring $I_2$-TCA complexes and the signal amplification through network cascading of the 3D long-range ordered framework. Finally, the conductance reaches a saturation of ~$1.27 \times 10^{-3}$ S at the strain level of 3.3%. To be more representative and compelling, over 100 samples are tested and all of them exhibit similar performance.

As re-plotted in Fig. 3b, the device current develops in three obvious stages with the increasing strains. When the device is bended below 2.5%, the device current is almost constant. Upon being bended between 2.5% and 3.3%, the device current increases exponentially with an ultrahigh gauge factor of 11,200 observed at 3.3% strain level. Once the applied strain exceeds 3.3%, the device current becomes constant and saturated. The minor conductance variation observed at the strain levels <2.5% and >3.3% can be ascribed to the bending-induced contact resistance increase at the probe/electrode interfaces, and/or the formation and propagation of cracks in the Au electrodes if exists[10–13], which do not influence the sensing performance of the $I_2$@CuTCA device. As the device resistance decreases (instead of

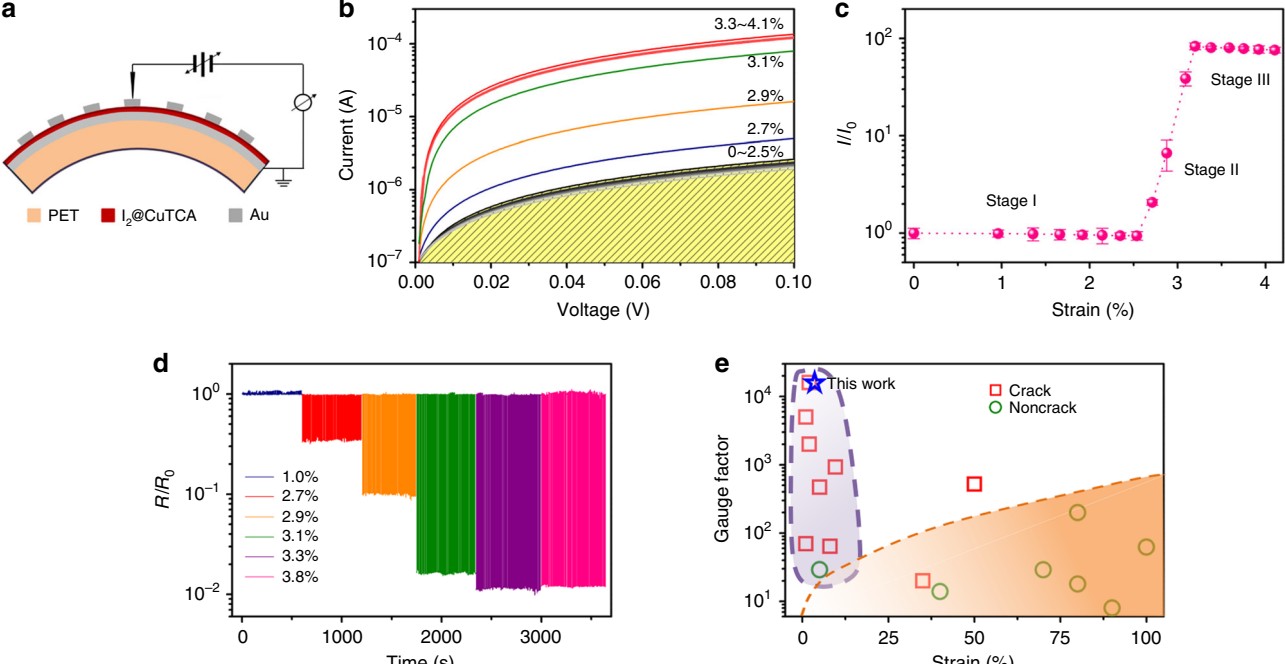

**Fig. 3** Electromechanical properties of the $I_2$@CuTCA nanofilm. **a** Schematic illustration of the bended device structure fabricated on PET substrate for electrical measurements. **b** Current–voltage curves of the Au/$I_2$@CuTCA/Au device under various bending strains of 0% to 4.1%. **c** Evolution of the device current as a function of the bending strains. The currents are read at 0.1 V and normalized to that measured at 0% strain. **d** Endurance of the Au/ $I_2$@CuTCA/Au device at the bending strains of 1.0%, 2.7%, 2.9%, 3.1%, 3.3%, and 3.8%, respectively. Device resistances are normalized to that measured at 0% strain. **e** Gauge factor comparison of the $I_2$@CuTCA device with other strain sensors documented in the literature[12–14,17,37,57–67]

increase as in the contact or crack resistance) in the responding range of 2.5% to 3.3%, as well as no crack appears in the MOF layer until the strain exceeds 4.3% (Supplementary Figure 7), the strain-responsive conductivity is convinced as an intrinsic property of the $I_2$@CuTCA nanofilms. The current–voltage and current–strain characteristics of the MOF devices are similar to the output and transfer curves of a conventional field-effect transistor, assembling the world's first two-terminal molecular analog of the strain-gated transistors[56]. The low hysteresis nature of the sensing curve in addition allows stable dynamic operations (Supplementary Figure 8). Endurance of the MOF devices were assessed at the strain of 2.7%, 2.9%, 3.1%, and 3.3%, respectively. At each level, the device was subjected to continuous bending operations of more than 1000 times, wherein the conductance responses of the device stay reproducible and reliable (Fig. 3c). The $I_2$@CuTCA device also carries mechanosensitivity and gauge factor comparable to the highest value of crack-type sensors documented in the literatures (Fig. 3d)[11–13,16,25,57–67], which make ultrasensitive, anti-jamming, and durable strain sensing possible with MOF materials.

**Strain-responding mechanism.** A better understanding of the ultrasensitive and anti-jamming strain sensing with $I_2$@CuTCA nanofilm is realized by correlating the local conductivity evolution with the microstructure variation of the iodine-doped MOF film through in-situ conductive atomic force microscopic (C-AFM) observation and X-ray diffractive analysis. In C-AFM observation, the topography of the $I_2$@CuTCA nanofilm and the mapping of its nanoscale conducting regions are recorded using a Pt/Ir-coated conductive AFM tip (Supplementary Figure 9). The localized current map is recorded at 0.1 V under various bending strains (Supplementary Figure 10). Overlying of the C-AFM current map on top of the MOF layer topography indicates that minor localized leakage conductive regions initially present at the grain boundaries of the flat particulate nanofilm (Fig. 4a) and remain almost unchanged as the bending strain increases to 2.5%

(Fig. 4b, c). This coincides with the non-responsiveness of the device conductance observed during macroscopic electrical measurements and can be explained in terms of strain release at grain boundaries. Owing to the presence of large amount of defects (e.g., missing of either organic linkers or metal ions, or both) and lack of sufficient chemical bonding between the neighboring grains, as well as the local low materials densities, the grain boundaries exhibit low mechanical strength and ease of deformation in comparison to the crystalline grains. Low level tensile strains can thus be released through the minor deformation of the grain boundaries without altering the microstructure of the MOF crystal grains, as is confirmed by the in-situ XRD analysis (Fig. 5a) demonstrating constant diffractive patterns of the MOF thin film in the strain range of 0% to 2.5%. Herein, the (222) peak signals of the MOF nanofilm are recorded as shown in Supplementary Figure 11. As the $I_2$@CuTCA crystals are free from structure and electronic property variations in this scenario, the device current will be still mainly arising from the minor leaking paths of the grain boundaries, resulting in overall non-responsive low conductivity of the device under small strain levels.

Further bending the sample may induce simultaneous shrinkage of the $I_2$@CuTCA particulates with the deformation of the grain boundaries, as is evidenced from the obvious shifts in the (222) peak $2\theta$ angle of ~1° when the strain increases from 2.5% to 3.3% (Fig. 5a). Being consistent with the MD simulation results (Fig. 2), it decreases the charge hopping distance between the neighboring $I_2$-TCA complexes inside the framework and increases the conductivity of the MOF crystals to much larger than that of the grain boundaries (Fig. 4d–f). Consequently, the conductive regions gradually extend from the edge of the grains into the interior parts of the MOF particulates and account for the macroscopic strain responsiveness of the device in the middle strain ranges. As the bending strain exceeds 3.3%, the relative low-strength grain boundaries will undergo significant yet still reversible deformation more easily when in comparison with the MOF crystals, which again helps to effectively release the tensile

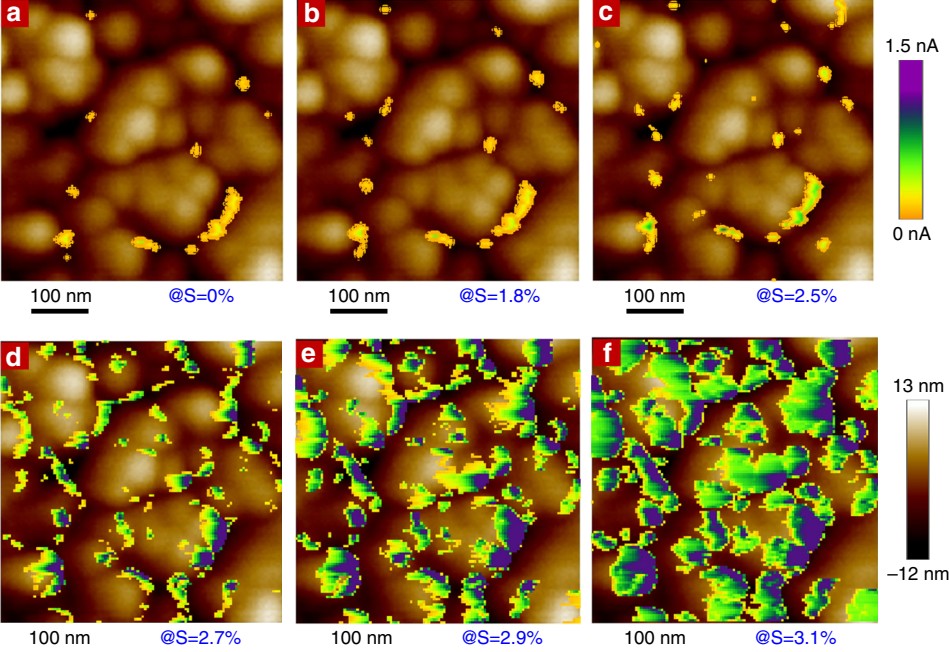

**Fig. 4** Microscopic current and topography maps of the $I_2$@CuTCA nanofilm. **a–c** With the bending strains under 2.5%, the conductive regions of the particulate metal-organic framework nanofilm merely appear at the grain boundaries. **d–f** When the bending strain exceeds 2.5%, the conductive regions gradually extend into the entire nanograins with their local conductivity increasing simultaneously and continuously

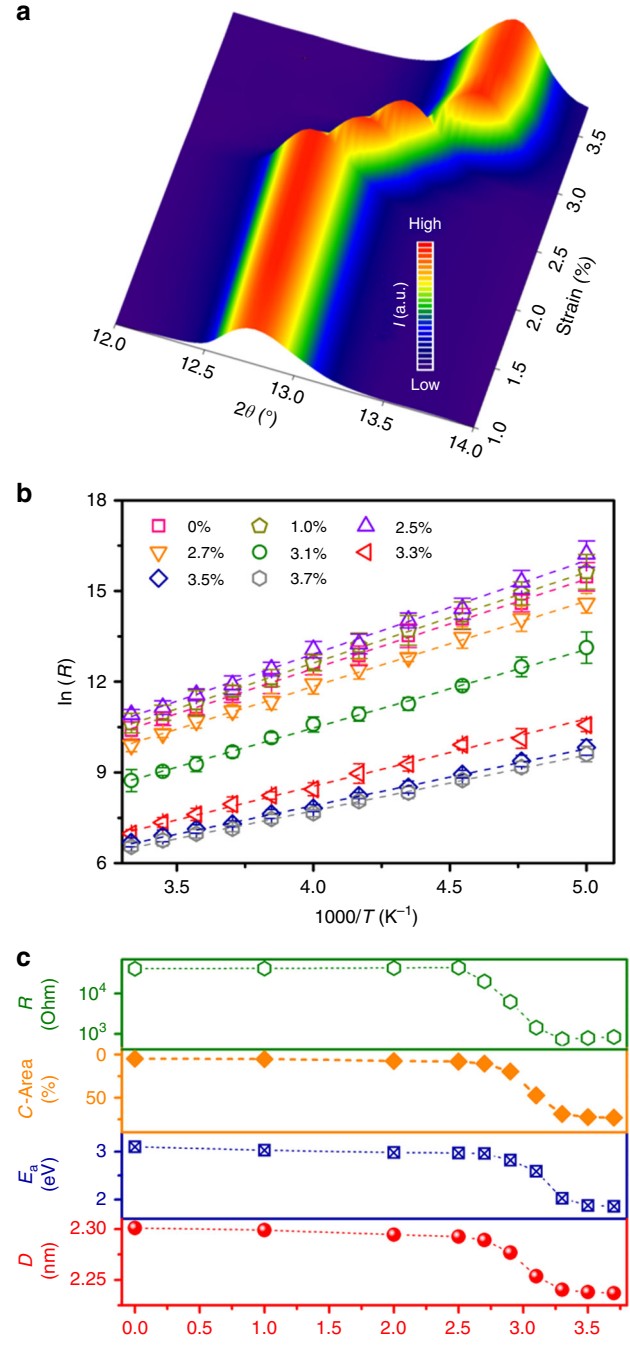

**Fig. 5** Strain-sensing mechanism of the $I_2$@CuTCA nanofilm. **a** X-ray diffraction pattern of the $I_2$@CuTCA nanofilm under bending strains of 1.0% to 3.5%. **b** Resistance-temperature characteritics of the MOF sensor under different bending strains. **c** Resistance–strain (R–S), conductive area–strain (C–Area–S), activation energy–strain ($E_a$–S) and 111-direction crystalline interplane spacing-strain (D–S) relationships of the $I_2$@CuTCA nanofilm

$(E_a/T)$ (Fig. 5b), as the bending strain increases from 0% to 4%. The reduction in $E_a$ is in good agreement with the bandgap decrease of 1.49 eV as derived through theoretical simulation. Figure 5c compares the lineshapes of the device resistance–strain, conductive area (counted from C-AFM data)–strain, and thermal activation–strain relationships with the (111) direction crystalline interplane distance-strain curve of the in-situ XRD results. As can be seen, all of the four curves exhibit similar evolution manner, convincing that the structure deformation-induced modulation of the inter-atom spacing between the $I_2$-TCA spots is responsible for the strain-sensing capability of the $I_2$@CuTCA nanofilms.

The strain-gating range of the MOF device can be further modulated by varying the iodine-doping level. As for common sense, increasing the doping level can both increase the charge-carrier concentration inside the MOF framework and lower the distance between the neighboring $I_2$-TCA complexes (thus increase the charge-carrier hopping probability and mobility). As such, hopping transport inside the $I_2$@CuTCA nanofilm can be effectively enhanced, leading to a device current increasing with the $I_2$ molecule/N atom ratio (Supplementary Figure 12a). Mechanical deformation in the more heavily doped MOF can result in more significant decrease in the hopping distance and thus faster increase in device conductivity. Consequently, the minimum responding strain of the $I_2$@CuTCA device decreases from 3.2% for sample with the $I_2$/N ratio of 0.13:1 to 2.5% for sample with the $I_2$/N ratio of 1.07:1 (Supplementary Figure 12b). Further increasing the doping level over saturation does not change the electromechanical performance of the device significantly. On the other hand, the responsiveness ($I/I_0$) of the $I_2$@CuTCA devices also increases with the increasing iodine levels. In accordance with the electromechanical performance of the macroscopic devices, the local conductivity of the MOF nanofilms demonstrate the same evolution manner, wherein the increasing doping level leads to more significant increase in the conductivity of the $I_2$@CuTCA nanograins (Supplementary Figure 10, 13, 14).

**Detection and recognition of human body motions**. The ultrasensitive, anti-jamming, and durable strain-sensing characteristics of the $I_2$@CuTCA device can be used to distinguish the human body motions of moderate muscle hyperspasmia from subtle swaying and vigorous exercising. As demonstration, a smart kneecap is designed to detect signals from the motion of our knee joints. The $I_2$@CuTCA sensor (on the $1\,cm \times 1\,cm$ square sample shown in the inset of Fig. 6a) is sewed onto a stainless steel annular chain that can convert the movement of knee joint into bending strains and transmit it to the sensor. Due to the 100 nm thickness and transparent nature of the MOF layer, the circular shape top Au electrode cannot be clearly differentiated from the bottom Au electrode by naked eyes. Then they are integrated with a 20 kΩ resistor and a 3 V button cell to form the strain detecting unit and mounted onto a commercial kneecap that is comfortable for daily wearing (Fig. 6a). By recording the change of voltage that falls onto the 20 kΩ resistor, the resistance (or conductance) of the $I_2$@CuTCA sensor can be real-time monitored with a bluetooth-enabled mobile phone. As such, the extent of bending strain sensed by the MOF device and thus the type of knee movement is recognized. For instance, anoetic leg swaying with tiny vibration of the knee joint during rest usually results in the strain of 4.5% in the kneecap. The strain sensed by the MOF device is much lower than 4.5% upon being transmitted through the stainless steel annular chain, which does not trigger the MOF sensor (Fig. 6b). This is different from a vibratory gyroscope based pedometer[68]. By monitoring the output voltage signal of the 20 kΩ resistor, a 0.7% deformation

strain of the film before the appearance of non-elastic deformation and/or cracking. As such, the geometrical and electronic structure of the $I_2$@CuTCA particulates stops to evolve (Fig. 5a) and the total conductive area of the MOF nanofilm stabilizes (Supplementary Figure 10), corresponding to the macroscopic device current saturation at the same strain level.

Interestingly, the activation energy for carrier hopping transport inside the iodine-doped MOF nanofilm changes from 3.13 eV to 1.88 eV according to the Arrhenius equation $\sigma \sim \exp$

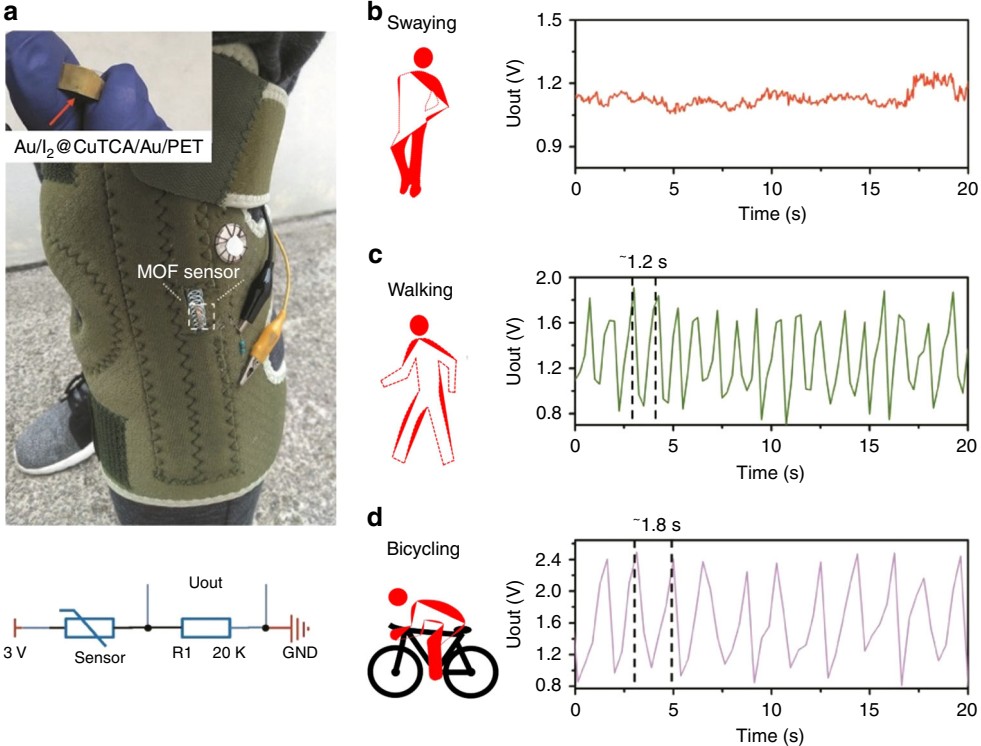

**Fig. 6** Human body motion recognition with the $I_2$@CuTCA strain sensor. **a** Schematic of the home-made smart kneecap with integrated MOF sensor, a 20 kΩ resistor (R1) and a button cell. Photo image of the sensor and equivalent circuit of the strain detecting unit are also plotted. Uout stands for the voltage falling on the resistor R1. **b–d** Strain-responding voltage output signals of the smart kneecap under leg swaying (**b**), walking (**c**), and bicycling (**d**) conditions, respectively

sensed by the MOF sensor is derived. In case of gentle walking and strenuous bicycling, the kneecap strain levels are ~20% and 28%, respectively. The corresponding voltage signals are obviously different in terms of spike intensity (1.8 V vs. 2.4 V) and duration (1.2 s vs. 1.8 s) (Fig. 6c, d), which reveal a strain of ~2.7% and 3.2% as sensed by the MOF sensor, respectively. By handling the output voltage signal through the following empirical algorithm, we could also estimate the calorie burning during exercises by

$$Ec = W \times D \times k \tag{1}$$

where $Ec$ is the total energy consumed in the unit of kcal, $W$ is the body weight in kg, $D$ is the total walking distance (km), and $k$ is an empirical constant of ~0.8214 for walking. During 1 h walking of ~6238 steps (with the step width of 0.6 m/step) for a 65 kg male adult, a total walking distance of 3.74 km and energy consumption of ~198 kcal/828 kJ is achieved (Supplementary Movie 2)[69], equivalent to 0.03 calories per walking step. This information is important for body fitness and medical health management.

## Discussion

In conclusion, ultrasensitive, anti-jamming, and durable strain-sensing capability is endowed to MOF-based molecular devices for the first time, arising from the structure deformation-induced modulation of charge-carrier hopping process. The highest gauge factor of 11,120 as reported so far and the middle-range responsiveness of the $I_2$@ CuTCA strain sensor may not only be used for remote health-monitoring purpose, but also help in infrastructure failure diagnosis and prevention of buildings, bridges, etc., in normal working scenario, as well as under natural disaster (e.g., hurricanes and earthquakes) conditions.

## Methods

**Sample preparation**. Au/PET substrates were prepared by sputtering a 5 nm buffer layer of titanium and subsequently a 200 nm layer of gold onto the pre-polished PET substrates at room temperature, in an E-beam evaporation chamber at a base pressure of ~$10^{-5}$ Pa. Then the as-prepared Au/PET substrates were placed in piranha solution ($H_2SO_4:H_2O_2 = 4:1$) for 30 s and rinsed with excess amount of deionized water to achieve Au-OH functionalization. Afterwards, the functionalized conductive substrates were sealed into a home-designed reactor for the solvothermal growth of CuTCA nanofilm through a modified LPE approach according to our previous work[44]. The substrates were soaked in a $50 \times 10^{-3}$ M ethanol solution of $Cu(CH_3COO)_2$ for 10 min. After removing the Cu salt solution, the substrate was rinsed with excess ethanol for 5 min, to remove the unreacted $Cu^{2+}$ ions, and blow-dried in the reactor with a nitrogen stream. Subsequently, $40 \times 10^{-3}$ M ethanol solution of $H_3$TCA was injected into the reactor and reacted with the $Cu^{2+}$ anchored substrate for another 10 min. After washing the sample again with ethanol adequately, the above procedure has been repeated for 50 times to grow 50 layers of $Cu_3(TCA)_2$ on the substrates. The entire preparation process was performed automatically without being exposed to the ambient atmosphere, avoiding potential contamination of the samples from the air. After MOF deposition, the samples were heated under reduced pressure at 100 °C for 30 min, to remove solvent ethanol molecules completely. Finally, 80 nm-thick top Au electrodes with a diameter of 100 μm were deposited onto the sample by sputtering Au target in pure argon atmosphere with a metal shadow mask. In order to obtain the iodine-doped MOF samples, the 1 cm × 1 cm × 100 nm CuTCA nanofilm deposited on Au/PET substrates was soaked in a 2 mL $I_2$/ethanol solution with the iodine concentration of 100 μM for 48 h, immediately after heating it under reduced pressure. The CuTCA nanofilms were also soaked in $I_2$/ethanol solutions with different concentrations of 5 μM, 10 μM, and 1 mM to obtained samples with various $I_2$ doping levels. Afterwards, the samples were heated at 80 °C for 1~2 h, to remove the residue solvent and free iodine molecules that do not form stable interaction with the $H_3$TCA ligand.

**Characterization**. The crystalline structure of the as-grown CuTCA nanofilms was investigated by glazing-incidence XRD technique (GIXRD, Bruker AXS, D8 Discover) using Cu-K α radiation. The incidence angle of the X-ray beam was fixed at 1° and the diffractive patterns were recorded with a step of 0.05° in the range of 5° to 90°. Background noises are screened from the raw diffractive data sets with the official Bruker/EVA 4.0 software. The thickness of the films was determined using field-emission scanning electron microscopy (Hitachi, S-4800) with 15 kV

accelerating voltage. Raman spectra were obtained with a Renishaw inVia Raman microscope. Core-level XPS of the MOF layers were monitored using a SHI-MADZU Axis Ultra Dld system. A monochromatic Al-K α X-ray source (1486.6 eV photons) was used at a constant dwell time of 100 ms. A pass energy of 80 or 40 eV was employed for the wide and core-level spectra scans, respectively. The X-ray source was run at a reduced power of 150 W (15 kV and 10 mA). The pressure in the analysis chamber was maintained at $10^{-8}$ Torr or lower during each measurement. The core-level signals were recorded at a photoelectron take-off angle (with respect to the sample surface) of 90°. All BEs were referenced to the C 1 s hydrocarbon peak at 284.6 eV. In curve fitting, the full width at half-maximum for the Gaussian peaks was maintained constant for all components in a particular spectrum. Surface elemental stoichiometries were determined from the XPS spectral area ratios and were reliable within ± 5%. The elemental sensitivity factors were calibrated using stable binary compounds of well- established stoichiometries. Background correction and peak fit features were performed to the data using CASA XPS software. Topography of the samples was recorded on a Leica DM2500 M optical microscope and a Bruker Dimension 3100 V (Veeco) scanning probe microscope, which is equipped with a Pt/Ir-coated conducting cantilever for C-AFM measurements. The $I$–$V$ characteristics of the Au/CuTCA/Au and Au/I$_2$@CuTCA/Au structures were measured on a Lakeshore probe station equipped with a precision semiconductor parameter analyzer (Keithley 4200) in dc sweeping mode. Bending of the devices to different levels was done with a home-made apparatus quantitatively, which is equipped with a pair of displacement sensors and computer program to automatically measure the distance between the two ends of the sample.

## Data availability

The authors declare that the main data supporting the findings of this study are available within the article and its Supplementary Information files. Extra data are available from the corresponding author upon request.

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

## Acknowledgements

We acknowledge the financial supports from, the National Key R&D Program of China (2017YFB0405604 and 2016YFA0201102), the National Natural Science Foundation of China (61722407, 61774161, 61674153, 51525103, and 11474295), the Natural Science Foundation of Zhejiang Province (LR17E020001), K. C. Wong Education Foundation (rczx0800), National Research Foundation, Prime Minister's Office of Singapore under the NRF Investigatorship (NRF2016 NRF-NRFI001-21), Singapore Ministry of Education (MOE2015-T2-2-060), and Academic Research Fund Tier 1 (RG107/15 and RG2/16). We thank Professor Dr Hongxin Yang and Dr Huali Yang from Ningbo Institute of Materials Technology and Engineering (CAS) for fruitful discussion.

## Author contributions

L.P., G.L., and R.-W.L. conceived the idea. L.P. prepared and characterized the sample. L.P. and J.S. conducted the electrical measurements. W.S. performed the MD simulation. L.P., W.R.L., Y.L., M.X., and S.L. designed smart kneecap and conducted electro-mechanical measurements. L.P., G.L., X.D.C., and R.-W.L. co-wrote the paper. All the authors discussed the results and commented on the manuscript.

## Additional information

**Competing interests:** The authors declare no competing interests.

