## [Peer Review File · Nature Communications]

Reviewers' comments:

Reviewer #1 (Remarks to the Author):

Liang Pang et al presented an interesting report for highly sensitive strain gauge sensor based on metal organic frameworks. They made organic nanolattice which can be deformable via strain ranging from 2.5% to 3.3% and demonstrated human motion detector. Manuscript is well written and data characterization is also nicely presented. I strongly recommend the publication of this manuscript in Nature Communications, after the authors make some following revisions.

[Major]

1. Is it possible controlling the strain-gate range? Changing the bond length with different materials or different ratio of Iodine dopant may make it possible. Was there any attempts for it? Otherwise any other ideas for the gate control?
2. Could you add the picture of how your sensor looks like and the how much dimensions it is?? Please show us photoimage of the device
3. Is there any morphological change after addition of Iodine? How large is the one MOF structures??
4. What is the thickness of the substrate? Under bending status, strains at outer covering and inner one are different. Especially, strain direction changes by "neutral plane" of the substrate. Therefore, substrate thickness is an important value to judge whether the neutral plane is in the PET layer or in the MOF. Otherwise, in case the neutral plane is in the MOF, the MOF may be under compressive strain. Is the property changed in other substrate, which have different Poisson's ratio?
5. (Line 142 & Fig2 a) Shrinkage by the external strain is quite confusing. It seems it's because of the Poisson's ratio and the figure2 (a) is the molecular structures on the cross section area which can be influenced by the Poisson's ratio. It would be better to clearly mention this information. If the shrinkage phenomenon under external strain is caused by another reason, please explain it.
6. The lack of scientific reasons explaining the out-of-scale nonresponsiveness is a lot regrettable. In line 80–83, it explains with the release of strain by the grain boundaries but this is not fully acceptable yet. Furthermore, even with the relaxation of the grain boundary it is rarely understandable for higher (3.3%<) strain.
7. Usually strain gauge sensors display normalized values like R/R_0 or V/V_0 . Please change all data in Fig. 3 with normalized ones.

[Minor]

1. I wonder how the bending strain was measured. It is mentioned that samples were quantitatively bent by the home-made apparatus (Fig s1). Then, is the bending strain a calculated value by the curvature? I don't think the apparatus can automatically give the bending strain and the curvature information as well. Furthermore, I believe the bending strain in this paper is the strain at the most outer covering since the MOF is way thinner than the PET substrate. If it's not, please mention the exact spot where the strain value is pointing out
2. Could you kindly indicate what is what in Figure 1a? For example, I'm confused what is yellow spheres (seems like copper?) and green twigs and red spheres (seems like iodine?)
3. (Fig 2d) Hope the 4% strain graph also has the band gap mark like the dotted lines at 0% strain graph.
4. Could you show us the algorithm it has used for the calories calculation?
5. (Fig 1a) Giving the structural formula of the TCA, CU after reaction would be better.
6. Fig. 4 is not well presented. Important information, strain % are too small and not be highlighted. Both scale bars also poorly presented. Please include minimum values in both.
7. (Fig 3d) Please refer to 'Dramatically Enhanced Mechanosensitivity and Signal-to-Noise Ratio of Nanoscale Crack-Based Sensors: Effect of Crack Depth. Adv. Mater., 28: 8130-8137. doi:10.1002/adma.201602425 '. It's about a crack sensor with the gauge factor of 16,000 in 2% strain.

Reviewer #2 (Remarks to the Author):

Li et al report on the fabrication of thin films of a Cu(II) MOF by using a bottom-up methodology. The film is then exposed to iodine to produce a nanostructured I₂@MOF film that displays very sensitive response to strain linked to changes in conductivity. This flexible strain-resistive film is incorporated into a device that permits to discriminate between different types of exercise based on different motion of knee joints. Overall, I recognise the effort of the authors of translating fundamental science into a functional device based on the preparation of strain-responsive films. I would be happy to recommend publication after a major revision that addresses important concerns on the scientific issues pivotal to the operational principles that regulate electrical changes in the film.

After reading the work, it is unclear to me what is the actual origin of the electrical conductivity in I₂@CuTCA films. In some paragraphs the authors ascribe this to:

1. I₂-MOF charge transfer pathways (In 74-76)
2. Semi-continuous tunnelling channel between adjacent I₂-TCA complexes (104-106)
3. They claim the tunnelling transport follows a thermally-dependent activated semiconducting model (236-237), not consistent with points 1 or 2.

The authors seem to use different concepts to accommodate the experimental findings without giving a clear idea of what is the origin of electrical conductivity in the film (tunnelling, band conduction). They even use DOS diagrams to explain changes in the electronic structure of the film with strain (not a single detail is provided that explains how these were calculated) that would only be reasonable for a band conduction mechanism.

Other points:

1. From my experience it is difficult to obtain nice diffraction data with a 100 nm thick film by using GIXDR unless performed with synchrotron radiation. Figure 1b shows very nice diffraction data with almost negligible noise and very sharp diffraction lines. How was this experiment performed?
2. The authors assume a 1:1 ratio of I₂ and TCA linkers for computational modelling. I cannot find any experimental evidence that support this claim in the paper.
3. The only evidence for suggesting charge transfer upon I₂ loading is Raman spectroscopy. I recommend complementary XPS and EPR studies to confirm this point.
4. (In 159) CuTCA nanofilm has an epitaxial particulate nature. It is certainly built from segregated crystalline particles but there is nothing that suggests an epitaxial growth.
5. The authors use local-contact conductive AFM experiments to correlate local conductivity changes with the changes to the microstructure of the film by application of strain. This study could be extended to different samples with variable microstructure (this can be varied quite easily with minor changes: concentration or soaking time in the fabrication step) for more general conclusions.
6. How is strain % quantified?
7. I miss relevant references linked to charge transport in MOF films like: Allendorf, M. D. *Science* 2014, 343 (6166), 66; Martí-Gastaldo, C. *Adv. Mater.* 2018, 55, 1704291; Coronado, E. J. *Am. Chem. Soc.* 2016, 138 (8), 2576 or Wenzel, W. *ACS Nano* 2016, 10 (7), 7085.
8. Based on the microstructure of the films presented it is more adequate to estimate peak to peak height rather than RMS to account for the suitability of the films for device fabrication.

Response to Reviewers' Comments and List of Revisions

Title: “Mechano-Regulated Metal-Organic Framework Nanofilm for Ultrasensitive and Anti-Jamming Strain Sensing”

Authors: Liang Pan, Gang Liu*, Wenxiong Shi, Jie Shang, Wan Ru Leow, Yaqing Liu, Ying Jiang, Shuzhou Li, Xiaodong Chen* and Run-Wei Li*

ALL THE CHANGES ARE MADE IN RED IN THE REVISED MANUSCRIPT

Response to Reviewers #1' Comments

General Comment:

Liang Pang et al presented an interesting report for highly sensitive strain gauge sensor based on metal organic frameworks. They made organic nanolattice which can be deformable *via* strain ranging from 2.5% to 3.3% and demonstrated human motion detector. Manuscript is well written and data characterization is also nicely presented. I strongly recommend the publication of this manuscript in Nature Communications, after the authors make some following revisions.

Response:

We thank the reviewer for the positive comment on our work. We have carefully considered all the reviewers' comments and revised the manuscript accordingly as described below.

Major Comment #1: Is it possible controlling the strain-gate range? Changing the bond length with different materials or different ratio of Iodine dopant may make it possible. Was there any attempts for it? Otherwise any other ideas for the gate control?

Response:

We thank the reviewer for his/her questions. Theoretically, either changing the bond length with different organic linkers or ratios of the iodine dopant may control the strain-gating range through modulating the hopping distance between the adjacent I₂-TCA complexes. According to the reviewer's suggestion, we prepared a series of I₂@CuTCA samples with different doping levels. By soaking the CuTCA nanofilm in I₂/ethanol solutions with different iodine concentrations of 1 mM, 100 μM, 10 μM, 5 μM and 0 μM, respectively, I₂@CuTCA nanofilms with the I₂ molecule/N atom ratio of 1.13:1, 1.07:1, 0.27:1, 0.13:1 and 0:1 are obtained (as estimated by XPS analysis). As discussed in the manuscript, increasing the doping level can both increase the charge carrier concentration inside the MOF framework, and lower the distance between the neighboring I₂-TCA complexes (thus increase the charge carrier hopping probability and mobility). As such, hopping transport inside the I₂@CuTCA nanofilm can be effectively enhanced, leading to a device current increasing with the I₂/N ratio (**Figure S11a**). Mechanical deformation in the more heavily doped MOF can result in more significant decrease in the hopping distance and thus faster increase in the device conductivity. Consequently, the minimum responding strain of the I₂@CuTCA

device decreases from 3.2% for sample with the I₂/N ratio of 0.13:1 to 2.5% for sample with the I₂/N ratio of 1.07:1 (**Figure S11b**). Further increasing the doping level over saturation does not change the electromechanical performance of the device significantly. On the other hand, the responsiveness (I/I_0) of the I₂@CuTCA devices also increases with the increasing iodine doping levels. In accordance with the electromechanical performance of the macroscopic devices, the local conductivity of the MOF nanofilms demonstrate the same evolution manner, wherein the increasing doping level leads to more significant increase in the conductivity of the I₂@CuTCA nanograins (**Figure S12 and S13**).

Related discussion has been added in the revised manuscript on page 15 line 7 to page 16 line 2.

Major Comment #2: Could you add the picture of how your sensor looks like and the how much dimensions it is? Please show us photoimage of the device.

Response:

We thank the reviewer for his/her suggestion. The photograph of the flexible Au/I₂@CuTCA/Au/PET device is shown in the inset of **Figure 6a** in the revised manuscript. The square sample is ~ 1 cm long × 1 cm wide × 175 μm thick, while a single Au/I₂@CuTCA/Au/PET device is 380 nm in thickness and 100 μm in diameter. Due to the hundred nanometer thickness and transparent nature of the MOF layer, the circular shape top Au electrode cannot be differentiated from the bottom Au electrode.

Related discussion has been added in the revised manuscript on page 6 line 11-13 and page 16 line 8-13.

Major Comment #3: Is there any morphological change after addition of Iodine? How large is the one MOF structures?

Response:

We thank the reviewer for his/her question. The *fcc*-structured CuTCA unit cell is 23.211 Å in all the a, b and c directions, wherein the use of relative large TCA linkers allows sufficient space for guest molecule infiltration. As such, neither does the crystalline structure nor the morphology of the MOF layer change upon doping with iodine, as shown in **Figure S1, S3 and S10** of the revised supplementary information.

Related discussion has been included in the revised manuscript on page 6 line 16-17, and page 7 line 9-10.

Major Comment #4: What is the thickness of the substrate? Under bending status, strains at outer covering and inner one are different. Especially, strain direction changes by “neutral plane” of the substrate. Therefore, substrate thickness is an important value to judge whether the neutral plane is in the PET layer or in the MOF. Otherwise, in case the neutral plane is in the MOF, the MOF may be under compressive strain. Is the property changed in other substrate, which have different Poisson's ratio?

Response:

We agree with the reviewer that the substrate thickness and position of the neutral

plane play an important role in determining the strain type of the flexible samples under bending conditions. In the present study, the thickness of the substrate is ~ 175 μm , which is much thicker than that of the Au/I₂@CuTCA/Au device (~ 380 nm). As such the neutral plane is definitely located in the PET substrate and the MOF layer experiences tensile strain upon being bended upwards as shown in **Figure S6**. On the other hand, the substrate Poisson's ratio may not change the in-plane strain direction that the MOF layer faces under bending conditions, since the parts of sample above the neutral plane are all experiencing tensile strains.

Related discussion is also added in the revised manuscript on page 8 line 11-21 and in the revised Supplementary Information on page 13 and 14.

Major Comment #5: (Line 142 & Fig2 a) Shrinkage by the external strain is quite confusing. It seems it's because of the Poisson's ratio and the figure2 (a) is the molecular structures on the cross section area which can be influenced by the Poisson's ratio. It would be better to clearly mention this information. If the shrinkage phenomenon under external strain is caused by another reason, please explain it.

Response:

We are sorry for making confusion when discussing the shrinkage of the MOF framework. As seen by the reviewer, **Figure 2a** shows the molecular structures of the I₂@CuTCA framework on the cross section area under increasing strains. Due to the positive Poisson's ratio, which is the usual case for most of the existing materials, the in-plane stretching of the sample upon upward bending will lead to shrinkage in the cross section plane of the MOF layer as shown in **Figure 2a**.

Related discussion is also added in the revised manuscript on page 8 line 11-21.

Major Comment #6: The lack of scientific reasons explaining the out-of-scale nonresponsiveness is a lot regrettable. In line 80~83, it explains with the release of strain by the grain boundaries but this is not fully acceptable yet. Furthermore, even with the relaxation of the grain boundary it is rarely understandable for higher ($3.3\% <$) strain.

Response:

We are sorry for not providing sufficient explanation on the release of strain by the grain boundaries and the consequent out-of-scale non-responsiveness of the MOF sensor devices. Due to the presence of large amount defects (e.g. missing of either organic linkers or metal ions, or both) and lack of sufficient chemical bonding between the neighboring grains, as well as the local low materials densities, the grain boundaries exhibit low mechanical strength and ease of deformation in comparison to the crystalline grains. Low level tensile strains can thus be released through the minor deformation of the grain boundaries without altering the microstructure of the MOF crystal grains, as is confirmed by the *in-situ* XRD analysis (**Figure 5a**) demonstrating constant diffractive patterns of the MOF thin film in the bending strain range of 0% to 2.5%. Since the I₂@CuTCA crystals are free from structure and electronic property variations, the device current will be still mainly arising from the minor leaking paths of the grain boundaries, resulting in overall non-responsive low conductivity of the

device under small strain levels. As the bending strain exceeds 3.3%, the relative low-strength grain boundaries will undergo significant yet still reversible deformation more easily when in comparison to the MOF nanocrystals, which again helps to effectively release the strain of the nanofilm before the appearance of non-elastic deformation and/or cracking. As such, the geometrical and electronic structure of the I₂@CuTCA particulates stops to evolve and the total conductive area of the MOF nanofilm stabilizes, corresponding to the macroscopic device current saturation at the same strain level.

Related discussion has been added in the revised manuscript on page 13 line 10-22 and page 14 line 10-17.

Major Comment #7: Usually strain gauge sensors display normalized values like R/R₀ or V/V₀. Please change all data in Fig. 3 with normalized ones.

Response:

We thank the reviewer for his/her suggestions. The device currents and resistances are changed to normalized values in **Figure 3b** and **3c** of the revised manuscript.

Minor Comment #1: I wonder how the bending strain was measured. It is mentioned that samples were quantitatively bent by the home-made apparatus (Fig S1). Then, is the bending strain a calculated value by the curvature? I don't think the apparatus can automatically give the bending strain and the curvature information as well. Furthermore, I believe the bending strain in this paper is the strain at the most outer covering since the MOF is way thinner than the PET substrate. If it's not, please mention the exact spot where the strain value is pointing out.

Response:

We thank the reviewer for his/her questions. The bending strain is indeed a calculated value. As discussed in the Supplementary Information, the bending strain experienced by the device is estimated as that encountered at the outer covering of the Au/I₂@CuTCA/Au/PET multilevel structure. As the thickness of the Au/I₂@CuTCA/Au device (380 nm in total) is much smaller than that of the PET substrate (175 μm), the strain across the entire Au/I₂@CuTCA/Au structure is almost uniform, and can be

expressed as $S = \frac{t_s}{L} \times \sqrt{\frac{6(L-D)}{L}}$, where t_s is the thickness of the PET substrate, L is

the original length of flat device and D is the distance between the two ends of the sample upon being bended (**Figure S6**). Distance D can be automatically measured through the home-made bending apparatus equipped with a pair of displacement sensors, which is controlled and read through a computer program as shown in **Figure S5**.

Detailed discussion has been added in the revised Supplementary Information on page 13 and 14.

Minor Comment #2: Could you kindly indicate what is what in Figure 1a? For example, I'm confused what is yellow spheres (seems like copper?) and green twigs

and red spheres (seems like iodine?)

Response:

We thank the reviewer for his/her suggestions. In **Figure 1a**, the cyan, white, blue and red twigs represent carbon, hydrogen, nitrogen and oxygen atoms, respectively. The yellow and red spheres stand for copper and iodine atoms, respectively.

Related descriptions are added in the Figure Legends of **Figure 1** and **2** in the revised manuscript.

Minor Comment #3: (Fig 2d) Hope the 4% strain graph also has the band gap mark like the dotted lines at 0% strain graph.

Response:

We thank the reviewer for his/her suggestion. The bandgap for I₂@CuTCA at both 0% and 4% strain levels are marked in **Figure 2d** in the revised manuscript.

Minor Comment #4: Could you show us the algorithm it has used for the calories calculation?

Response:

We thank the reviewer for his/her suggestion. Generally, the total calorie burning during exercises can be estimated through the following empirical equation

$$Ec \text{ (kcal)} = \text{Body Weight (kg)} \times \text{Distance (km)} \times k \quad (1)$$

where Ec is the total energy consumed and k is an empirical constant of ~ 0.8214 for walking. During 1 hour walking of ~ 6238 steps (with the step width of 0.6 m/step) for a 65 kg male adult, a total walking distance of 3.74 km and energy consumption of around 198 kcal/828 kJ is achieved. The energy consumption during exercises can be also estimated through internet tools, such as that cited in Ref 69 with the link <https://theblueroom.bupa.com.au/healthier/be-active/energy-burned>.

Related discussion is included in the revised manuscript on page 17 line 2-8.

Minor Comment #5: (Fig 1a) Giving the structural formula of the TCA, CU after reaction would be better.

Response:

We thank the reviewer for his/her suggestion. The molecular formulas of H₃TCA (C₂₁H₁₅NO₆) and CuTCA (Cu₃(C₂₁H₁₂NO₆)₂) are included in the revised manuscript on page 5 line 21 and page 6 line 1, as well as in **Figure 1a**.

Minor Comment #6: Fig. 4 is not well presented. Important information, strain % are too small and not be highlighted. Both scale bars also poorly presented. Please include minimum values in both.

Response:

We thank the reviewer for his/her suggestions. The font size of strain% has been increased and highlighted in light blue color for a better illustration in **Figure 4** of the revised manuscript. The minimum values for both the current and morphology scale bars are added in the figures also.

Minor Comment #7: (Fig 3d) Please refer to ‘Dramatically Enhanced Mechanosensitivity and Signal-to-Noise Ratio of Nanoscale Crack-Based Sensors: Effect of Crack Depth. Adv. Mater., 28: 8130-8137. doi:10.1002/adma.201602425 ‘. It’s about a crack sensor with the gauge factor of 16,000 in 2% strain.

Response:

We thank the reviewer for reminding us to include the above reference Adv. Mater. 2016-28-8130 for the comparison of the present sensor performance with those documented in the literatures. The reference has been cited as Ref 67 in the revised manuscript.

Response to Reviewers #2' Comments

General Comment:

Li et al report on the fabrication of thin films of a Cu(II) MOF by using a bottom-up methodology. The film is then exposed to iodine to produce a nanostructured I₂@MOF film that displays very sensitive response to strain linked to changes in conductivity. This flexible strain-resistive film is incorporated into a device that permits to discriminate between different types of exercise based on different motion of knee joints. Overall, I recognise the effort of the authors of translating fundamental science into a functional device based on the preparation of strain-responsive films. I would be happy to recommend publication after a major revision that addresses important concerns on the scientific issues pivotal to the operational principles that regulate electrical changes in the film.

Response:

We have carefully considered the reviewer's comments and revised the manuscript accordingly as described below.

Comment #1: After reading the work, it is unclear to me what is the actual origin of the electrical conductivity in I₂@CuTCA films. In some paragraphs the authors ascribe this to:

1. I₂-MOF charge transfer pathways (ln 74-76)
2. Semi-continuous tunnelling channel between adjacent I₂-TCA complexes (104-106)
3. They claim the tunnelling transport follows a thermally-dependent activated semiconducting model (236-237), not consistent with points 1 or 2.

The authors seem to use different concepts to accommodate the experimental findings without giving a clear idea of what is the origin of electrical conductivity in the film (tunnelling, band conduction). They even use DOS diagrams to explain changes in the electronic structure of the film with strain (not a single detail is provided that explains how these were calculated) that would only be reasonable for a band conduction mechanism.

Response:

We are sorry for making confusion about the conduction and mechano-regulated conduction of the I₂@CuTCA nanofilms. As discussed in detail in the revised manuscript, the incorporation of I₂ molecules can form charge transfer complex with the TCA linkers, which leads to oxidative doping with increased hole concentration. However, due to the large size of the TCA linker and small size of the I₂ molecule, continuous through-bond conduction pathway may not be established by charge transfer interaction, as in the case of TCNQ@HKUST-1. On the other hand, the distance between the adjacent I₂-TCA complexes (8.42 Å) is still larger than the van der Waals distance (**Figure 2b**), indicating that the through-space band conduction is unlikely to occur either. Nevertheless, temperature-dependent I-V measurement of the I₂@CuTCA device shows thermally-activated conductivity, suggesting that hopping mechanism may account for its semiconductive nature. When the stretching strain is applied and increases, the I₂-TCA to I₂-TCA distance decreases gradually and reaches

6.38 Å at the strain level of 4% (**Figure 2c**). The shortened spatial distance between the hopping sites can thus increase the hopping probability and consequently the charge carrier mobility. An accompanying lowering of the energy bandgap of 1.49 eV upon deformation is also demonstrated, which can be ascribed to the enhanced electronic communication between the adjacent I₂-TCA charge transfer complexes (**Figure 2d**). Consequently, the local deformation of MOF lattice in response to the external strains enhance the charge transport across the nanofilm, and engender strain-responsive conductivity of the Au/I₂@CuTCA/Au/PET device.

In general, charge transfer interaction increases the carrier concentration of I₂@CuTCA, while charge hopping is the main conduction mechanism. The calculated DOS diagrams shown in **Figure 2d** is used to evaluate the band structure variation of the MOF upon deformation, and is not contrary to the hopping mechanism. The calculation details were provided in the revised supplementary information on page 3. The detailed discussion on the conduction and strain-responsiveness are added in the revised manuscript on page 9 line 1 to page 10 line 12.

Other Comment #1: From my experience it is difficult to obtain nice diffraction data with a 100 nm thick film by using GIXDR unless performed with synchrotron radiation. Figure 1b shows very nice diffraction data with almost negligible noise and very sharp diffraction lines. How was this experiment performed?

Response:

We thank the reviewer for his/her questions. Background noises of the as-obtained XRD spectrum were screened from the raw data sets with the official Bruker/EVA 4.0 software. To be more accurate, the original diffractive pattern of the CuTCA nanofilm without noise screening is also shown below. As can be seen, clear (222) and (444) peaks of the CuTCA framework are observed, as that plotted in **Figure 1c** in the manuscript.

Related description is included in the revised manuscript on page 19 line 14-15.

Other Comment #2: The authors assume a 1:1 ratio of I₂ and TCA linkers for computational modelling. I cannot find any experimental evidence that support this claim in the paper.

Response:

We thank the reviewer for his/her comment. In order to confirm the occurrence of charge transfer interaction between the infiltrated iodine molecules and the TCA linkers, as well as the doping level of the I₂@CuTCA nanofilms, X-ray photoelectron spectroscopic (XPS) measurements were conducted to monitor the oxidative states of the N atoms in the TCA linker. As shown in **Figure S2a** and **S2b**, the core-level N1s XPS spectrum of the I₂ doped CuTCA nanofilm reveals a noticeable appearance of new nitrogen species with the binding energy of ~ 399.2 eV and 400.2 eV, respectively, which can be ascribed to the presence of neutral N atoms at the high oxidative state (e.g. imine) and positively charged N atom upon charge transfer between the I₂ molecule and MOF framework. By calculating XPS spectral area ratio of the iodine and nitrogen species in the I₂@CuTCA nanofilm that is obtained by soaking the pristine CuTCA sample in iodine solution with the I₂ concentration of 100 μM, a 1.07:1 ratio of the I₂ molecule/N atom is obtained (**Figure S2c** and **S2d**). Thus, during MD simulation, we use a 1:1 ratio of I₂ and TCA linker for computational modelling.

Related discussion has been included in the revised manuscript on page 7 line 14-21 and in the revised supplementary information on page 8 and 9. The details for XPS analysis is also added in the METHOD section on page 19 line 18 to page 20 line 10.

Other Comment #3: The only evidence for suggesting charge transfer upon I₂ loading is Raman spectroscopy. I recommend complementary XPS and EPR studies to confirm this point.

Response:

We thank the reviewer for his/her suggestions. As recommended by the reviewer, we conducted X-ray photoelectron spectroscopic (XPS) measurements to confirm the charge transfer interaction between the infiltrated I₂ molecules and the TCA linker. Please refer to our response to the previous comment for details.

Other Comment #4: (ln 159) CuTCA nanofilm has an epitaxial particulate nature. It is certainly built from segregated crystalline particles but there is nothing that suggests an epitaxial growth.

Response:

We thank the reviewer for his/her comments. As reported first by Christof Wöll (JACS 2007-129-15118) and reviewed later by Roland A. Fischer (Chem. Rev. 2012-112-1055) and others (Phys. Chem. Chem. Phys. 2008-10-7257, Nature Mater. 2009-8-481), the stepwise layer-by-layer adsorption of components from the liquid on the substrate surface under well-controlled experimental conditions can lead to MOF nanofilm with preferentially oriented crystalline structure. The synthesis of such high quality MOF film is termed as Liquid-Phase Epitaxy (LPE) in the above references. In

this study, we adopted and modified the LPE approach to deposit CuTCA nanofilm on (111)-Au/PET substrate. Well following that of the gold substrate (**Figure S10**), the as-obtained CuTCA layer also exhibits preferentially (111) crystalline orientation with (222) and (444) Bragg peaks in the XRD pattern of **Figure 1d**. Herein the term “epitaxial growth” is refereeing to the synthesized of thin films (in particulate nature for CuTCA) that resemble the crystalline orientation of the substrate, instead of producing thin film in single crystal forms. In order to avoid misunderstanding, the term “epitaxial” is removed from page 6 line 14, and is replaced by “preferentially-oriented crystalline” on page 10 line 15.

Other Comment #5: The authors use local-contact conductive AFM experiments to correlate local conductivity changes with the changes to the microstructure of the film by application of strain. This study could be extended to different samples with variable microstructure (this can be varied quite easily with minor changes: concentration or soaking time in the fabrication step) for more general conclusions.

Response:

We thank the reviewer for his/her suggestions. According to the reviewer’s suggestion, we prepared a series of I₂@CuTCA samples with different doping levels. By soaking the CuTCA nanofilm in I₂/ethanol solutions with different iodine concentrations of 1 mM, 100 μM, 10 μM, 5 μM and 0 μM, respectively, I₂@CuTCA nanofilms with the I₂ molecule/N atom ratio of 1.13:1, 1.07:1, 0.27:1, 0.13:1 and 0:1 are obtained (as estimated by XPS analysis). Increasing the doping level can lower the distance between the neighboring I₂-TCA complexes, while mechanical deformation in the more heavily doped MOF can result in more significant decrease in the hopping distance and thus faster increase in the device conductivity. As such, the C-AFM current map of the I₂@CuTCA nanofilms with higher doping level demonstrate more obvious increase in the local conductivity of the MOF nanograins as shown in **Figure S10, S13 and S14**.

Related discussion has been added in the revised manuscript on page 15 line 7 to page 16 line 2.

Other Comment #6: How is strain % quantified?

Response:

We are sorry for not providing the details of estimating the bending strains applied to the device. As shown in **Figure S5** in the revised Supplementary Information, the bended surface of the device is generally assumed as an ideal part of a perfect circle at all bending radius (*Adv. Mater.* 2013-25-5425; *Flexible Electronics: Materials and Applications*, Springer-Verlag, New York 2009), while the slight deviation from the perfection may lead to some acceptable error for bending radius estimation. The strain S induced on the surface of the bended device can be estimated from following the equations:

$$S = \frac{(t_D + t_s)(1 + 2\eta + \chi\eta^2)}{2R(1 + \eta)(1 + \chi\eta)} \quad (1)$$

$$\chi = Y_D / Y_S \quad (2)$$

$$\eta = t_D / t_S \quad (3)$$

where t_D and Y_D are the thickness and Young's modulus of the device, t_S and Y_S are the thickness and Young's modulus of the PET substrate, and R is the curvature radius of the sample upon being bended. Since the thickness of the device is much smaller ($t_D=80+100+200+5=385$ nm) than that of the PET substrate ($t_S \approx 7$ mil, or $175 \mu\text{m}$), equation (1) can be simplified as

$$S \approx t_S / 2R \quad (4)$$

while the strain across the entire device (e.g. at the top and bottom surface of the MOF layer) is almost uniform, which is also due to the much thinner thickness of the device in comparison to that of the substrate. Generally, the radius (R) of the curvature can be estimated from the following equations:

$$L = \theta \times R \quad (5)$$

$$\sin\left(\frac{\theta}{2}\right) = \frac{D/2}{R} \quad (6)$$

where L is the original length of flat device, θ is the angle of the curvature and D is the direct sample between the two sides of the device upon being bended. Distance D can be automatically measured through the home-made bending apparatus equipped with a pair of displacement sensors as shown in **Figure S5**. With Taylor expansion the strain of the curved sample can be expressed as:

$$S = \frac{t_S}{L} \times \sqrt{\frac{6(L-D)}{L}} \quad (7).$$

Related discussion has been added in the revised Supplementary Information on page 13 and 14.

Other Comment #7: I miss relevant references linked to charge transport in MOF films like: Allendorf, M. D. Science 2014, 343 (6166), 66; Martí-Gastaldo, C. Adv. Mater. 2018, 55, 1704291; Coronado, E. J. Am. Chem. Soc. 2016, 138 (8), 2576 or Wenzel, W. ACS Nano 2016, 10 (7), 7085.

Response:

We thank the reviewer for reminding us to include the above references for the discussion of charge transport in MOF thin films. The references have been cited as Ref 30-33 in the revised manuscript.

Other Comment #8: Based on the microstructure of the films presented it is more adequate to estimate peak to peak height rather than RMS to account for the suitability of the films for device fabrication.

Response:

We thank the reviewer for reminding us to provide the peak-to-peak height of the MOF nanofilm. As estimated from the AFM measurements shown below, the

maximum peak-to-peak height of the as-fabricated 100 nm thick MOF film is ~ 10 nm, which is suitable for device fabrication.

Related description has been included in the revised manuscript on page 6 line 10-11.

REVIEWERS' COMMENTS:

Reviewer #1 (Remarks to the Author):

The authors have responded to reviewer comments very well. I believe that the manuscript is suitable for publication in Nature Communication after revising additional minor issue.

1. Abstract cannot show novelty of the work. In line 21 (page 2), only electromechanical property is mentioned. Please more address uniqueness.
2. In line 78, gauge factor should have exact strain information. "in the strain range between 2.5% to 3.3%" is not the specific strain value.
3. Figure 3a shows more broad range signal (0~4.1%) than Fig. 3c. Thus, in figure 3c, please include at least two more data. I think R/R₀ with less than 2.5 and higher than 3.3 % are required.
4. Figure 6. needs applied strain information. For three different actions, swaying, walking and bicycling, it is necessary to explain how much strains are applied.

Response to Reviewers' Comments and List of Revisions

Title: "Mechano Regulated Metal Organic Framework Nanofilm for Ultrasensitive and Anti Jamming Strain Sensing"

Authors: Liang Pan, Gang Liu^{*}, Wenxiong Shi, Jie Shang, Wan Ru Leow, Yaqing Liu, Ying Jiang, Shuzhou Li, Xiaodong Chen^{*} and Run-Wei Li^{*}

ALL THE CHANGES ARE MADE IN "TRACK-CHANGE" FEATURE

Response to Reviewers #1' Comments

General Comment:

The authors have responded to reviewer comments very well. I believe that the manuscript is suitable for publication in Nature Communication after revising additional minor issue.

Response:

We thank the reviewer for the positive comment on our work. We have carefully considered the reviewer's comments and revised the manuscript accordingly.

Comment #1: Abstract cannot show novelty of the work. In line 21 (page 2), only electromechanical property is mentioned. Please more address uniqueness.

Response:

We thank the reviewer for his/her comments and suggestions. We have further emphasized the novelty of the work in the abstract on page 2 and shorten the abstract according to the editor's requirements as well.

Comment #2: In line 78, gauge factor should have exact strain information. "in the strain range between 2.5% to 3.3%" is not the specific strain value.

Response:

We thank the reviewer for his/her suggestion. The gauge factor 11120 is observed at the strain level of 3.3%. We added this data in the revised manuscript on page 4 line 21 to page 5 line 1, and on page 12 line 4-5..

Comment #3: Figure 3a shows more broad range signal (0~4.1%) than Fig. 3c. Thus, in figure 3c, please include at least two more data. I think R/R₀ with less than 2.5 and higher than 3.3 % are required.

Response:

We thank the reviewer for his/her suggestion. R/R₀ at 1% and 3.8% strain levels are added in the revised manuscript.

Comment #4: Figure 6. needs applied strain information. For three different actions, swaying, walking and bicycling, it is necessary to explain how much strains are applied.

Response:

We thank the reviewer for his/her suggestion. Generally, the strains induced on the kneecap during swaying, walking and bicycling are 4.5%, 20% and 28%, respectively. Upon being transmitted to the MOF device with the stainless steel annual chain, the strains sensed by the sensor are lower. By monitoring the voltage falling on the fixed resistor and thus the current flowing through the MOF device, the strains sensed by the MOF devices are calculated to be 0.7%, 2.7% and 3.2% during swaying, walking and bicycling, respectively. Related discussion is added in the revised manuscript on page 17 line 9-19.